# REPLACING LOSS FUNCTIONS AND TARGET REPRESENTATIONS FOR ADVERSARIAL DEFENSE

**Sean Saito & Sujoy Roy**
SAP Asia, Singapore
{`sean.saito, sujoy.roy`}@sap.com

## ABSTRACT

Recent works have shown that neural networks are susceptible to adversarial data, despite demonstrating high performance across various tasks. Hence, there is a growing need to develop techniques that make neural networks more robust against attacks given their increasingly frequent applications in real-life use cases. In this work, we propose simple techniques for adversarial defense, namely: (1) changing the loss function from cross entropy to mean-squared error, (2) representing targets as codewords generated from random codebooks, and (3) using an autoencoder to filter noisy logits before the final activation layer. Our experiments on CIFAR-10 using the DenseNet model have shown that these techniques can help prevent targeted attacks as well as improve classification accuracy on adversarial data generated in a white-box or black-box setting.

## 1 INTRODUCTION

In this paper we investigate the problem of adversarial attacks on image classification systems. Our attack scenario is the white/black-box setting where the attacker can submit classification jobs to an image classification model (which takes an image and outputs the probabilities over different image categories). The attacker may (white-box) / may not (black-box) be aware of the specific machine learning architecture used to generate the model. In the case of a black-box scenario, the attacker uses another model (which can, in the worst case, be considered to be the same as the original model) to help generate adversarial examples.

We propose the following approaches in dealing with adversarial examples.

- Train the model with mean-squared error (MSE), rather than cross-entropy
- Replace the traditional one-hot representation for targets with non-deterministically generated codebooks
- Add an autoencoder layer (that has been pre-trained using adversarial examples) before the final softmax layer to filter noisy logits

## 2 BACKGROUNDS

The goal of an attack is to construct inputs that cause a model to make erroneous predictions. These inputs are referred to as *adversarial examples*. This work implements two common approaches to generating adversarial examples, namely the one-step gradient-based approach and the iterative approach. The former relies on taking the gradient of the loss function of the model to determine the perturbation added to the original data. The fast gradient sign method (FGSM), developed by Goodfellow et al. (2014), is one such attack:

$$X_{adv} = X + \epsilon * sign(\nabla_X J(X, Y_{correct}))$$

where $X_{adv}$ is the adversarial sample of $X$, $\epsilon$ is a scale parameter which determines the magnitude of perturbations, $J$ is the loss function of the model. Iterative methods, like the basic iterative method (BIM), repeatedly apply the above process (Kurakin et al. (2016)).

## 3   METHODOLOGY

This work investigates three modifications to the traditionally evaluated adversarial scenarios (DenseNet with CIFAR-10 dataset), namely changing the loss function, changing the target representation using codebooks and adding a denoising autoencoders to remove adversarial noise.

### 3.1   TRAINING USING MSE

We replace the cross-entropy loss function with mean-squared error with one-hot representation for targets. Details are elaborated in section 4.2.

### 3.2   CHANGING TARGET REPRESENTATIONS

We replace the one-hot target representations of the ground-truth with codewords from a codebook. For FGSM, a targeted attack looks like:

$$X_{adv} = X - \epsilon * sign(\nabla_X J(X, Y_{target}))$$

This technique can successfully subvert whitebox and targeted attacks as long as the attacker does not know $Y_{target}$ for a particular instance of the model. We generated these codebooks by sampling $n$ vectors from the uniform random distribution $U(-1, 1)^d$, where $n$ is the number of classes. Moreover, we replace the final softmax layer of the model with a tanh activation. For inference, we find the codebook target with the smallest Euclidean distance to the output.

### 3.3   DENOISING AUTOENCODER

We add a autoencoder before the final softmax layer of the DenseNet model. The autoencoder is separately trained to map the logits produced from adversarial inputs to logits produced from the original samples. Hence the autoencoder acts as a denoising filter. Details of the experiments are in section 4.4.

## 4   EXPERIMENTS

We report experimental results using the CIFAR-10 dataset to evaluate the effectiveness of the proposed approaches. All adversarial images are generated using only the test set consisting of $10,000$ images.

### 4.1   SETUP

We use the DenseNet model (Huang et al. (2017)), which has recently produced state-of-the-art results on several image datasets. We employ both one-step gradient-based methods (FGSM) and iterative methods (BIM), to generate adversarial images. For the BIM attack, we set the number of iterations to 10, which is the default value in the cleverhans library (Nicolas Papernot (2017)). We generate random codebook with each codeword having length $d = 128$. Codebooks with $d$ between 32 and 1024 gave similar results.

### 4.2   COMPARISON OF CROSS-ENTROPY AND MSE MODELS

We compare a DenseNet model trained on cross-entropy (CE) loss to one trained on MSE, both using one-hot target representation, with adversarial examples generated using a CE or MSE based model. The model trained using MSE performs the best in terms of classification accuracy, under adversarial examples generated by FGSM and BIM using both CE and MSE (first three rows of Table 1 and first two rows of Table 2).

"FGSM:CE–One-Hot:MSE" stands for adversarial example generated using FGSM with a CE loss based model applied to ('–') a classification model with one-hot target representation trained using MSE loss.

Table 1: Accuracies on Adversarial Images Generated by FGSM

| Approaches | $\epsilon$ | | | | | | |
|---|---|---|---|---|---|---|---|
| | 0.01 | 0.02 | 0.04 | 0.06 | 0.08 | 0.1 | 0.2 |
| FGSM:CE–One-Hot:CE | 73.88 | 56.04 | 39.79 | 32.70 | 28.63 | 26.40 | 17.89 |
| FGSM:CE–One-Hot:MSE | 89.11 | 82.25 | 69.27 | 58.68 | 51.59 | 44.84 | 23.76 |
| FGSM:MSE–One-Hot:MSE | 93.32 | 89.78 | 77.15 | 65.86 | 58.00 | 51.48 | 27.75 |
| FGSM:CE–Random:MSE | 89.46 | 84.91 | 74.99 | 64.32 | 54.92 | 46.49 | 24.90 |

Table 2: Accuracies on Adversarial Images Generated by BIM

| Approaches | $\epsilon$ | | | | | |
|---|---|---|---|---|---|---|
| | 0.005 | 0.01 | 0.015 | 0.02 | 0.04 | 0.06 |
| BIM:CE–One-Hot:CE | 83.45 | 67.08 | 54.24 | 45.70 | 29.98 | 26.61 |
| BIM:CE–One-Hot:MSE | 92.52 | 90.72 | 88.14 | 85.27 | 73.51 | 67.69 |
| BIM:CE–Random:MSE | 91.28 | 89.46 | 87.43 | 84.91 | 74.99 | 69.79 |

### 4.3 Target Representation to address Targeted Attacks

The last row of Table 1 and Table 2 present the performance of the DenseNet model using a random codeword representation instead of a one-hot representation. Note that the performance is similar to a one-hot MSE approach but is clearly better than a CE approach.

We generated targeted attacks using FGSM for the one-hot MSE and random codebook models under the assumption that the attacker has knowledge of the target representations. We measured the success rate, or the rate at which the model returns an attacker-desired prediction. Compared to the one-hot CE and MSE models, the random codebook model has a lower success rate even for large values of $\epsilon$. This justifies the use of an alternative target representation.

### 4.4 Using a Denoising Autoencoder

We compared classification accuracies on adversarial examples generated with FGSM using one-hot MSE models with and without a denoising autoencoder. The autoencoder has layers of dimensions [456, 8192, 1024, 8192, 456], where 456 is the shape of the logits before the softmax layer. We observe that for large values of $\epsilon$, the autoencoder helps increase performance significantly.

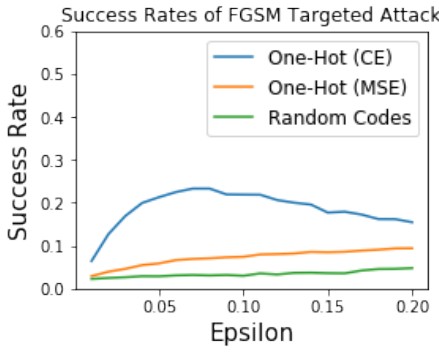

Figure 1: Targeted Attacks

Figure 2: Using a Denoising Autoencoder

## 5 Conclusion

We have proposed several techniques which can improve model resistance to FGSM and BIM attacks. While a theoretical analysis of these results requires additional investigation, we hope this work can help uncover further insight on making neural networks more secure.

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
