# OpenReview forum: "Replacing Loss Functions And Target Representations For Adversarial Defense"
_ICLR.cc/2018/Workshop — Reject_

### Official Review · AnonReviewer3 · 2018-02-23

**Rating:** 5
**Confidence:** 4

**Review:**

The authors propose 3 new methods for making deep networks robust to adversarial attacks. In particular, the authors propose (1) training with MSE error, (2) employing a random codebook and (3) leveraging an autoencoder to regularize the logits. All 3 methods led to improved robustness in the classifier to adversarial perturbations.

The authors show gains in adversarial robustness by using the prescribed techniques.

Pros:
- Each of the methods led to notable increased robustness to FGSM and BIM adversarial attack.

Cons:
- The authors do not report the accuracy of the new models on a 'vanilla' test dataset, so we do not know if there altered training procedures lead to poorer cross-validated models.
- Many details of the auto-encoder and codebook method not included in the paper. Additionally, the motivation for each approach is not entirely clear.
- The authors do not compare the fidelity of this method to other baseline approaches for adversarial defense.
- The paper lacks a clear motivation in terms of the suggested techniques.

---

### Official Review · AnonReviewer1 · 2018-03-06

**Rating:** 4
**Confidence:** 4

**Review:**

The paper presents three approaches to better defend an image model against adversarial attacks.
The first one is to replace the loss from cross entropy (CE) to mean squared error (MSE).
Results on adversarial attacks show better defense with this loss. Sadly, no results are shown to see the potential weakening of the model on normal images. It is known that MSE is usually worse than CE in these cases.
The second approach consider using a different code than one-hot for classification, and supposes this remains a secret and makes it hard to attack. Using output codes is a well-known robustness technique, called ECOC, and should be referred to. Once again, no result are provided on clean images.
The third proposal has been suggested in many other papers (for instance MagNet).

Overall, I found the paper not good enough for acceptance: it proposes 3 ideas, none of which are properly compared nor justified, and there's a lack of context from existing literature.

---

### Official Review · AnonReviewer2 · 2018-03-10
**Some new ideas**

**Rating:** 6
**Confidence:** 2

**Review:**

This paper presents three new techniques for defending neural network based image classifiers against adversarial images:
	- training with MSE loss rather than cross entropy loss
	- replacing one-hot label representation with randomly chosen binary codes for each label
	- add an auto-encoder layer before the final softmax to filter noise at this layer that the adversarial images may generate

Pros:
	- The defense techniques are novel as far as I know
	- All three defenses improve performance against the basic FGSM attack

Cons:
	- The techniques do improve the performance, but the gains are not as significant as existing defenses
	- They didn't compare to any other baseline defenses making it difficult to see how their model compares to existing work
	- They don't compare to adversarial samples generated from a model using their defense, except in one case.  Notably they don't report results on BIM for adversarial samples generated from any of their defenses
	- Surprisingly, when the adversarial images are generated using the MSE model, the MSE defense performs even better than when the adversarial images are generated using the CE model.  This result is surprising, and a bit suspect, so I would have liked to see an explanation of why this is happening (or at least an admission that they don't understand what's happening here).  This is particularly important given that this is the most important result in the paper

So overall, I think the paper has some novel ideas which have not yet been properly validated.  Given the goals for the workshop, to present new but not fully polished work, I think this puts it above the bar for acceptance.

---

### Decision · Program_Chairs · 2018-03-20
**ICLR 2018 Workshop Acceptance Decision**

**Decision:**

Reject

**Comment:**

Based on the reviews, this paper has not been accepted for presentation at the ICLR workshop. However, the conversation and updates can continue to appear here on OpenReview.